# The Prognostic Reliability of Lymphovascular Invasion for Patients with T3N0 Colorectal Cancer in Adjuvant Chemotherapy Decision Making

**DOI:** 10.3390/cancers14122833

**Published:** 2022-06-08

**Authors:** Hayoung Lee, Seung-Yeon Yoo, In Ja Park, Seung-Mo Hong, Seok-Byung Lim, Chang Sik Yu, Jin Cheon Kim

**Affiliations:** 1Department of Surgery, Asan Medical Center, University of Ulsan College of Medicine, Seoul 05505, Korea; d180359@amc.seoul.kr; 2Pathology Center, Seegene Medical Foundation, Seoul 133847, Korea; sypathology@mf.seegene.com; 3Department of Pathology, Asan Medical Center, University of Ulsan College of Medicine, Seoul 05505, Korea; 4Department of Colon and Rectal Surgery, Asan Medical Center, University of Ulsan College of Medicine, Seoul 05505, Korea; sblim@amc.seoul.kr (S.-B.L.); csyu@amc.seoul.kr (C.S.Y.); jckim@amc.seoul.kr (J.C.K.)

**Keywords:** adjuvant chemotherapy, colorectal cancer, lymphovascular invasion, recurrence, survival

## Abstract

**Simple Summary:**

This retrospective analysis evaluated the prognostic implications of lymphovascular invasion (LVI) associated with adjuvant chemotherapy in 1634 patients with pT3N0 colorectal cancer. Extensive pathologic review and dual immunohistochemical (IHC) staining with CD31 and D2-40 were undertaken in a subset of 242 patients to determine the reliability of LVI as a prognostic factor. The diagnosis of LVI and PNI changed in 82 (33.9%) and 61 (25.2%) patients, respectively, after central pathologic review (mean follow up duration, 50 (1–114) months). Five-year recurrence-free survival (RFS) and overall survival (OS) rates were 92% and 94.8%, respectively. Before and after pathologic review, LVI was not associated with OS but was associated with RFS after reviewing patients with pT3N0 colorectal cancer. In this patient cohort, the prognostic implications of LVI may have been underrecognized when using hematoxylin and eosin staining slides only for pathologic diagnoses, possibly leading to low recurrence prediction rates.

**Abstract:**

Lymphovascular invasion (LVI) is a high-risk feature guiding decision making for adjuvant chemotherapy. We evaluated the prognostic importance and reliability of LVI as an adjuvant chemotherapy indicator in 1634 patients with pT3N0 colorectal cancer treated with curative radical resection between 2012 and 2016. LVI and perineural invasion (PNI) were identified in 382 (23.5%) and 269 (16.5%) patients, respectively. In total, 772 patients received adjuvant chemotherapy. The five-year recurrence-free survival (RFS) and OS rates were 92% and 94.8%, respectively. Preoperative obstruction, PNI, and positive margins were significantly associated with RFS and OS; however, adjuvant chemotherapy and LVI were not. Pathologic slide central reviews of 242 patients using dual D2-40 and CD31 immunohistochemical staining was performed. In the review cohort, the diagnosis of LVI and PNI was changed in 82 (33.9%) and 61 (25.2%) patients, respectively. Reviewed LVI, encompassing small vessel invasion, lymphatic invasion, and large vessel invasion, was not an independent risk factor associated with OS but was related to RFS. The prognostic importance of LVI and adjuvant chemotherapy was not defined because LVI may be underrecognized in pathologic diagnoses using hematoxylin and eosin staining slides only, leading to low recurrence rate predictions. Using LVI as a guiding factor for adjuvant chemotherapy requires further consideration.

## 1. Introduction

In 2020, colorectal cancer was ranked third highest in terms of incidence and second highest in terms of mortality worldwide [1]. Guidelines for treatment and surveillance have accumulated over a long period and experience [2,3]. A combination of fluorouracil–oxaliplatin chemotherapy in the adjuvant setting provides a 30% relative risk reduction in disease recurrence and significantly improves overall survival (OS) [4,5]. While there is a consensus concerning adjuvant chemotherapy as standard treatment for stage III colon cancer, the benefits of adjuvant treatment for patients with lymph node-negative colorectal cancer remain controversial, particularly in stage IIA, which is T3N0 cancer [6,7,8]. Recent studies based on the National Cancer Database have reported on the OS benefit of adjuvant chemotherapy for patients with IIA colon cancer and high-risk prognostic features; however, the number of patients who received adjuvant chemotherapy was <50% and their performance status was not reported [6,9].

High-risk prognostic features such as preoperative obstruction, perineural invasion (PNI), lymphovascular invasion (LVI), <12 lymph nodes retrieved, poor differentiation, and resection margin involvement have been used when deciding whether adjuvant chemotherapy is indicated for patients with pT3N0 colorectal cancer [2,3]. Among the risk factors, LVI and PNI are well known and common pathologic high-risk features for oncologic outcome [10,11,12]. In particular, LVI has been considered a possible predictor of occult lymph node metastasis [10,12]. However, an LVI diagnosis depends on the observer and has frequently been considered contentious. Hematoxylin and eosin (H&E) staining has been routinely used to identify LVI, which has been denoted as vascular invasion regardless of the type of vessel involved (either lymphatic or blood vessel involvement). The College of American Pathologists (CAP) redefined LVI to enhance clarity [13] and identified small vessel invasion, including lymphatic and blood vessels without a muscle layer, as LVI and distinguished LVI from large vessel invasion in separate reports. However, the diagnosis of LVI remains challenging and time-consuming for pathologists. Additional staining methods, including elastic tissue staining, podoplanin, and other immunohistochemical (IHC) stains, have been used to distinguish between lymphatic and blood vessels, resulting in changes to the diagnosis [14,15]. This change in diagnostic methods has led to interobserver variability, resulting in a change in diagnosis in >20% of cases. An evaluation of the pathologic high-risk features that are vulnerable to diagnostic change is needed to determine whether they are reliable in guiding decision making for adjuvant chemotherapy.

This study aimed to evaluate the prognostic importance of high-risk features, especially LVI, in terms of the administration of adjuvant chemotherapy for patients with pT3N0 colorectal cancer.

## 2. Materials and Methods

### 2.1. Patients, Treatment, and Surveillance

From January 2012 to December 2016, we retrospectively analyzed data from our institutional database concerning 1634 patients with pT3N0 colorectal cancer who had undergone curative surgical resection at Asan Medical Center, Seoul, South Korea. We excluded patients with synchronous malignancy in other organs, preoperative chemoradiotherapy, familial adenomatous polyposis, hereditary nonpolyposis colorectal cancer, and patients who had undergone extended resection. Patients who could not be assessed during the pathologic stage were also excluded (Figure 1). This study was approved by the relevant institutional review board, and the requirement for informed consent was waived (2017-0955).

Adjuvant chemotherapy was recommended for patients with pT3N0 colorectal cancer showing one of the following high-risk features: the presence of preoperative obstruction that contained both endoscopic obstruction and clinically total obstruction, inadequate lymph node examination (<12 lymph nodes retrieved), the presence of LVI, PNI, poor histologic differentiation, and resection margin involvement. The microsatellite instability (MSI) status of the tumor was also evaluated.

Patients underwent a standardized postoperative follow-up that included physical examination, serum carcinoembryonic antigen (CEA) evaluation, laboratory tests with a complete blood cell count, liver function assessment, and plain chest radiographs every six months during the five years following surgery. They also underwent abdominal and pelvic computed tomography (CT) evaluations every 6 months. Colonoscopy was performed within the first year after surgery and then once every two or three years. If patients had preoperative obstruction, colonoscopy was performed within the six months following surgery. Recurrence-free survival (RFS) was defined as the interval between the date of resection of the primary tumor and the date of recurrence. OS was defined as the duration from the date of resection to any cause of death. The mean follow-up duration was 50 (1–114) months.

### 2.2. Pathologic Examination

The original pathologic reports followed a routine diagnosis protocol based only on H&E staining that included PNI, LVI, MSI status, number of retrieved lymph nodes, and resection margin status. These reports declared LVI while disregarding whether the invasion took place in lymphatic, small blood, or large vessels because the study period preceded that of the abovementioned CAP guidelines.

From the overall cohort, we identified patients receiving surgical resection during 2014 as a review cohort. Results of PNI and LVI after review were defined as the reviewed PNI (rPNI) and the reviewed LVI (rLVI), respectively. This cohort underwent a detailed central pathologic slide review that included PNI and its location, the presence of mucin, tumor grade, budding, resection margin status, and LVI. In addition to routine H&E staining, dual IHC for D2-40 and CD31 was performed for the differential diagnosis of lymphatic vs. small vessel invasion (Figure 2). Briefly, 4 μm thick tissue sections were deparaffinized and rehydrated through immersion in xylene and a graded ethanol series. Endogenous peroxidase was blocked through incubation in 3% H2O2 for 10 min, followed by heat-induced antigen retrieval. Immunohistochemical labeling was performed using an automatic stainer (Benchmark XT; Ventana Medical Systems, Tucson, AZ, USA), in line with the manufacturer’s protocol. In brief, sections were incubated at room temperature for 32 min with primary antibody against D2-40 (M3619, 1:100; Dako, Glostrup, Denmark) and then washed. Additional AP-conjugated mouse monoclonal CD31 (EP78, 1:800; Cell Marque, CA, USA) was used for dual CD31–D2-40 labeling. An UltraView AP Magenta Detection Kit (Ventana Medical Systems) was used for magenta chromogen, and the OptiView DAB Detection Kit (Ventana Medical Systems) was used for the brown chromogen. Immunolabeled sections were lightly counterstained with hematoxylin, dehydrated in ethanol, and cleared in xylene. The immunolabeled slides were reviewed by two pathologists. CD31 was labeled with magenta color and D2-40 was labeled with brown color, respectively. When tumor cells were surrounded by CD31 (red labeling)-labeled endothelial cells, the invasion was considered as venous invasion (Figure 2D). Lymphatic invasion was confirmed as the presence of tumor cells within the lymphatic space surrounded by D2-40-positive lymphatic endothelia (brown labeling; Figure 2B) [14]. Reviewed LVI (rLVI) was then described according to CAP guidelines, and categorized as lymphatic invasion, small vessel invasion, or large vessel invasion. Small vessel invasion was defined as a tumor deposit in or over the thin-walled lumen such as in lymphatic vessels or capillaries, and large vessel invasion as a tumor invasion in or beyond the endothelium structure with smooth muscle layer.

### 2.3. Statistical Analysis

Patients were categorized into two groups (one group received adjuvant chemotherapy and one group did not) to evaluate the association between risk factors and RFS and OS in both settings. Independent sample Student’s t tests were used to compare continuous variables, and chi-square or Fishers’ exact tests were used for categorical variables. We also used a chi-square test to evaluate the degree of discrepancy in LVI or PNI between results obtained before (original reports) and after (review reports) the slide reviews using dual IHC staining. A logistic regression test was used to evaluate the association between the clinicopathological risk factors and recurrence.

OS and RFS survival curves were plotted using the Kaplan–Meier method and compared using a log-rank test. A Cox regression test was performed to assess the prognostic effect of the risk factors on RFS and OS. The results were considered statistically significant with a *p* value < 0.05. Data analysis was performed using SPSS software (version 21.0; IBM Statistics, Armonk, NY, USA).

## 3. Results

### 3.1. Clinicopathological Characteristics of Patients with pT3N0 Colorectal Cancer

The clinicopathological features of the overall and review cohorts are shown in Table 1. Of 437 preoperative obstructions identified in the overall cohort, 351 (80.3%) were found to be endoscopic, and 86 (19.7%) were complete symptomatic obstruction. Circumferential margin involvement was observed in 11 of 12 patients with resection margin involvement in the overall cohort.

There was no significant difference in average age, sex ratio, tumor location, and other risk factors except for preoperative obstruction between the review and the overall cohorts. Preoperative obstruction was more frequent in the review cohort than in the overall cohort (33.9% vs. 26.6%, *p* = 0.028).

Of 351 patients with a single high-risk feature in the adjuvant chemotherapy group in overall cohort, 179 patients had LVI as the single high-risk feature. However, there were significantly more patients with high-risk features in the adjuvant chemotherapy group, and 38.1% of patients with high-risk features did not receive adjuvant chemotherapy (Table 2).

### 3.2. Recurrence, Oncologic Outcomes, and Associated Factors in the Overall Cohort

In the overall cohort, 116 (7.1%) patients experienced recurrence. The most common recurrence site was the lungs (*n* = 43, 37.1%), followed by the liver (*n* = 39, 33.6%), peritoneal seeding (*n* = 16, 13.8%), and lymph nodes (*n* = 11, 9.5%); local recurrence was identified in 10 (8.6%) patients.

The five-year RFS and OS rates were 92% and 94.8%, respectively. RFS was compared according to the indication for adjuvant chemotherapy in patients with high-risk features. Adjuvant chemotherapy administration was not associated with RFS (Figure 3A), but five-year OS significantly improved in patients treated with adjuvant chemotherapy (*p* < 0.001; Figure 3B).

Preoperative obstruction, PNI, and margin involvement were confirmed as factors associated with RFS in the multivariate analysis. Aging, preoperative obstruction, poorly differentiated histology, PNI, and margin involvement were confirmed as associated factors with a shorter OS rate. However, adjuvant chemotherapy and LVI were not associated with RFS or OS (Table 3).

### 3.3. Changes in the Diagnosis of LVI and PNI, and Oncologic Outcomes in the Review Cohort

In the review cohort, a total of 109 patients had large or small blood vessel invasion, or lymphatic invasion, which we defined as rLVI. In total, 82 (33.9%) patients had a change of LVI diagnosis following the central review (Table 4). The PNI status also changed in 61 (25.2%) patients via a central review of the slides without any additional staining for PNI detection (Table 4).

For the review cohort, the five-year RFS and OS rates were 94.3% and 93%, respectively. The five-year RFS rate did not differ according to the administration of adjuvant chemotherapy in patients with rLVI and rPNI (Figure 4).

The rLVI was the only factor associated with RFS in the univariate analysis (*p* = 0.043). In the review cohort, age, rPNI, and preoperative obstruction were found to be risk factors related to a lower OS. For both RFS and OS rates, adjuvant chemotherapy was not found to be an associated factor (Table 5).

## 4. Discussion

In this study, adjuvant chemotherapy was not found to be associated with RFS and OS in patients with pT3N0 colorectal cancer when considering high-risk features that indicated adjuvant chemotherapy. In addition, diagnoses of LVI and PNI, which are common high-risk features used to guide adjuvant chemotherapy, changed after additional staining or detailed central pathologic review.

Adjuvant chemotherapy is recommended based on its effect on oncologic outcomes. High-risk features suggested as clinicopathological features with prognostic importance are used to determine whether adjuvant chemotherapy administration is needed for patients with pT3N0 colorectal cancer. Representative guidelines suggest that poorly differentiated tumors, perforation/obstruction, inadequate LN acquisition, LVI, and PNI are high-risk prognostic features [2,3]. In addition to clinicopathologic high-risk features, patient factors such as comorbidity, age, performance status, and willingness to undergo treatment have also been considered important determinants for adjuvant chemotherapy administration in patients with pT3N0 colorectal cancer because the benefits of undergoing adjuvant chemotherapy have not been consistently defined.

Some studies have reported improved RFS or OS rates with adjuvant chemotherapy in patients with stage II colorectal cancer [16,17,18,19]. However, other studies have reported that complications and the harmful nature of chemotherapy can exceed its benefits for such patients [8,20,21,22]. One recent meta-analysis did not advocate chemotherapy because although it reported an RFS improvement rate of 5%, this was not a significant finding, and the improvement in OS in patients with stage II colorectal cancer was also not found to be significant [23]. In this study, adjuvant chemotherapy did not show a significant improvement in terms of RFS or recurrence reduction. This lack of significance might be attributable to the high five-year RFS rate in patients with T3N0 cancer (92%), which could be difficult to increase with the use of chemotherapy. Indeed, adjuvant chemotherapy was administered regardless of recommendations in many cases. The number of patients who received adjuvant chemotherapy in those with high-risk features was 61.9%, which is likely to have influenced outcomes. Considering that rLVI was associated with RFS in the review cohort, underdetection of pathologic high-risk features would be associated with an adjuvant chemotherapy recommendation.

In multivariate analysis, adjuvant chemotherapy alone was not found to be statistically significant as an associated factor with OS; however, the OS rate was significantly higher in the adjuvant chemotherapy group. Although we did not evaluate performance status, comorbidity, or postoperative complications, which are likely to interfere with adjuvant chemotherapy receipt, patients in the adjuvant chemotherapy group were significantly younger. Therefore, OS improvement in the adjuvant chemotherapy group may have been related to a favorable general status of patients able to receive adjuvant chemotherapy.

Some studies have analyzed the prognostic value of certain high-risk features and followed the effect of chemotherapy according to these features. The results, however, have been variable. Some studies have shown that LVI is a significant prognostic factor for either RFS or OS [11,24]. In contrast, other studies have reported no effect of LVI on RFS or OS [11,25]. Zhang et al. analyzed the effect of adjuvant chemotherapy in consideration of several risk factors and found no significant effect of adjuvant chemotherapy in patients with LVI, especially in terms of OS [26,27]. In this study, we could not detect a prognostic value for LVI in terms of RFS and OS in the overall cohort. However, LVI was associated with RFS in univariate analysis in the review cohort after central pathologic review. The lack of prognostic importance for LVI on RFS/OS found in the overall cohort may have been due to underdetection of LVI.

Despite this variability, indications for adjuvant chemotherapy continue to be based on high-risk features. However, patients’ demographic features and characteristics such as performance status, comorbidities, anxiety, age, and accessibility to health care are also considered factors for adjuvant chemotherapy indication because of the uncertain benefit of adjuvant chemotherapy in pT3N0 colorectal cancer. In this study, 772 (47.2%) patients received adjuvant chemotherapy. Although the adjuvant chemotherapy group had significantly more patients with high-risk features (73.2%), 325 (38.1%) patients in the nonadjuvant chemotherapy group also had high-risk features. In contrast, 31.8% of the adjuvant chemotherapy group had no high-risk features. This poor adherence to the guidelines for adjuvant chemotherapy may have been because the oncologic benefit of adjuvant chemotherapy has not yet been established.

Our results showed that adjuvant chemotherapy would be beneficial in oncologic outcomes in some patient subsets; however, careful consideration should be made concerning which factors are reliable and which factors should be included in the criteria for adjuvant chemotherapy administration in patients with pT3N0 colorectal cancer. The subjectivity and diagnostic variability of LVI assessment explain why LVI features frequently in debates on this matter [28,29,30]. Although the CAP does not recommend a particular staining method, additional IHC staining is suggested to improve the diagnosis given limitations using conventional H&E staining. These limitations usually include difficulty in differentiating between lymphatic invasion and blood vessel invasion. Therefore, some authors have emphasized the use of immunohistochemistry for vessel invasion diagnosis to improve the accuracy of LVI diagnosis [14,31]. Liang et al. reported that the false-positive rate of H&E staining used for LVI identification was 9.1% and that the false-negative rate was 12.6% using additional podoplanin and CD34 staining [32]. In an effort to achieve a more accurate diagnosis, Kingston et al. compared conventional H&E with some IHC staining such as elastic van Gieson, CD31, and CD34, and reported significant improvement in identifying vascular invasion [14].

Of the LVI diagnoses, >30% were changed in the review cohort and, finally, 45% of the review cohort showed rLVI. The original pathologic diagnosis was made only with H&E staining slides. Therefore, it might have been difficult to differentiate between retraction artifact and LVI (Figure 2A). In this study, two gastrointestinal pathologists performed a central pathology review using additional dual CD31-D2-40 immunolabeling and detected more cases with LVI present. Our results are similar to those of several previous studies that have reported detection of more foci of venous or lymphatic invasion using additional CD31 or D2-40 IHC staining, respectively, compared with conventional H&E staining only [14,30]. In the review cohort, rLVI was associated with RFS but not with OS, which differed from the overall cohort in which LVI was not found to be a risk factor for RFS or OS. This result may have been due to rLVI including large vessel invasion as part of its definition. In the overall cohort, large vessel invasion was not included in LVI, as it was considered an independent pathologic feature of LVI. Although blood vessel invasion has been reported as an independent predictor of oncologic outcomes [33,34,35], we found no association between independent large vessel invasion with either RFS or OS even after an extensive review of the pathologic high-risk features.

In our study, the diagnosis of PNI, an important high-risk feature in colorectal cancer, changed less often than the diagnosis of LVI (25.2% vs. 33.9%, respectively). PNI was found to be a significant prognostic factor for recurrence, RFS, and OS in our overall cohort. rPNI was also associated with poorer outcomes in the review cohort. Underdetection of PNI in the original pathology diagnoses could be explained by interobserver disagreement in relation to defining PNI. Therefore, establishing a more precise definition of PNI for colorectal cancers is required. PNI has been reported to have prognostic importance in oncologic outcomes [36,37]. Knijn et al. reported a strong association between PNI and local recurrence, highlighting its prognostic strength [36]. Yang et al. stated that patients with stage II colorectal cancer and PNI had similar outcomes compared with patients with stage III disease, implicating the importance of postoperative chemotherapy in patients with PNI [37]. PNI needs to be better evaluated in terms of its prognostic importance in colorectal cancer including the pT3N0 stage and clinical application.

This study had some limitations. We could not review all of the overall cohort’s pathologic slides. We reviewed those from 2014, and this may have limited our outcome analysis. However, patients’ clinicopathological characteristics did not differ between the overall and review cohorts, and the review cohort was representative of the overall cohort. Furthermore, the reasons for chemotherapy administration in 245 (31.8%) patients without any risk factors were unclear. Likewise, we did not categorize the reasons why 87 patients did not receive adjuvant chemotherapy when they had >2 risk factors. Finally, differences following chemotherapy regimens were not evaluated in this study.

## 5. Conclusions

Adjuvant chemotherapy did not improve RFS in pT3N0 colorectal cancer in this study regardless of high-risk features. LVI, which is the most common pathologic high-risk feature, was not found to be associated with lower RFS or OS rates in the overall cohort, but it was negatively associated with RFS after changing the diagnosis via a central pathologic review with additional dual CD31 and D2-40 IHC staining. Therefore, the reliability of LVI as a prognostic factor, specifically in patients with T3N0 potentially beginning chemotherapy, needs to be carefully considered. More detailed consensus on staining and classification in pathology diagnosis, and on the prognostic value of LVI, is warranted.

## Figures and Tables

**Figure 1 cancers-14-02833-f001:**
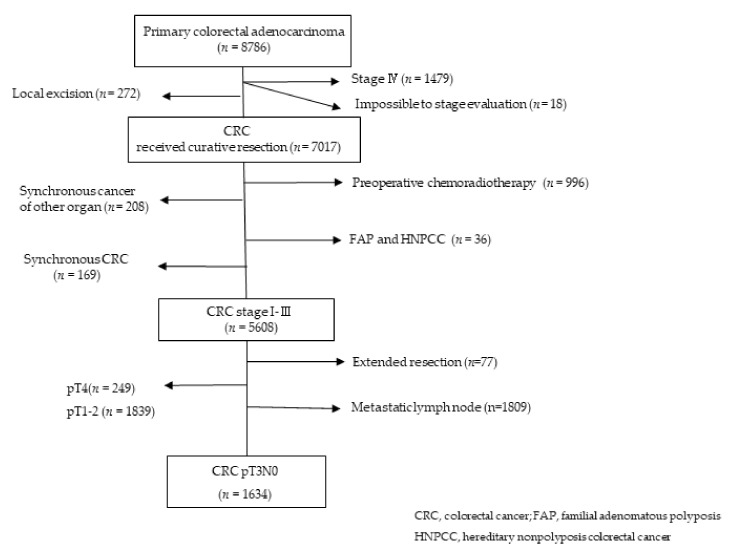
Inclusion and exclusion criteria for the overall cohort.

**Figure 2 cancers-14-02833-f002:**
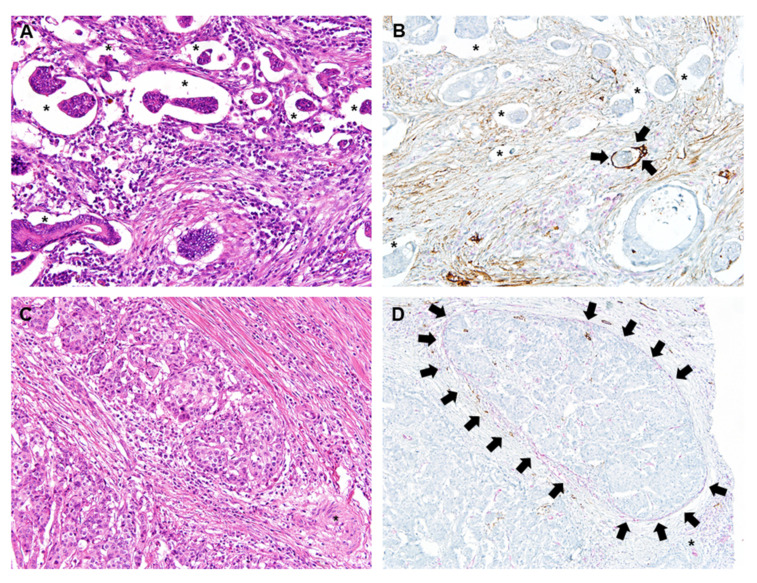
Representative images of venous and lymphatic invasion with original H&E staining and dual CD31-D2-40 immunolabeling. (**A**) Identifying lymphatic or venous invasion on H&E staining is extremely difficult due to retraction artifact (asterisks) making a space surrounding the cancer cells. (**B**) Dual CD31 (magenta)-D2-40 (brown) immunolabeling highlights cancer cell infiltration in lymphatic space (arrows) among numerous retraction artifacts (asterisks). (**C**) Identifying venous invasion on H&E staining is extremely difficult in this case. An unpaired artery (asterisk) is observed. (**D**) Dual CD31 (magenta)-D2-40 (brown) immunolabeling highlights venous invasion (arrows) of cancer cells. Venous endothelial cells show magenta color (CD31 labeling). Magnification of all images, ×200.

**Figure 3 cancers-14-02833-f003:**
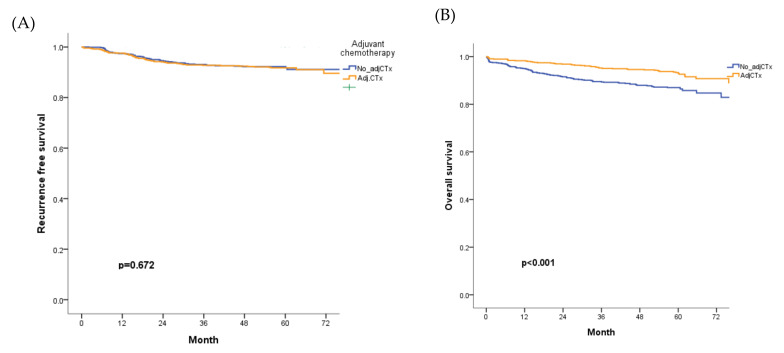
RFS and OS according to the administration of adjuvant chemotherapy in the overall cohort. (**A**) RFS did not differ according to the administration of adjuvant chemotherapy, (**B**) OS significantly improved for patients who received adjuvant chemotherapy Adj CTx; adjuvant chemotherapy; OS, overall survival; RFS, recurrence-free survival.

**Figure 4 cancers-14-02833-f004:**
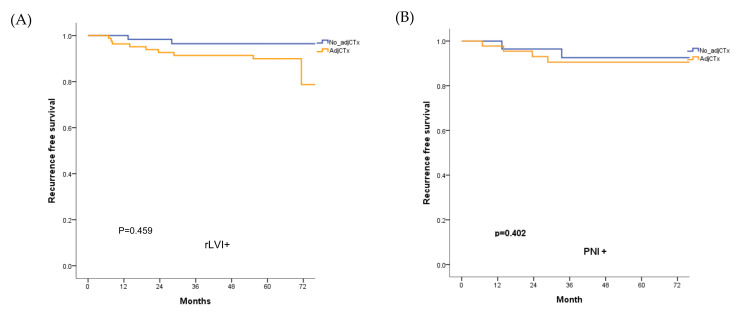
RFS according to the administration of adjuvant chemotherapy in patients with (**A**) rLVI and (**B**) rPNI in the review cohort. The RFS rate did not improve through the addition of adjuvant chemotherapy adj CTx; adjuvant chemotherapy; RFS, recurrence-free survival.

**Table 1 cancers-14-02833-t001:** The clinicopathological characteristics of patients with pT3N0 colorectal cancer in the overall and review cohorts.

	Overall Cohort (*n* = 1634)	Review Cohort (*n* = 242)
Age, mean ± SD, years	63 ± 12	62 ± 12
Sex		
Male	971 (59.6)	136 (56.2)
Female	663 (40.4)	106 (43.8)
Location		
Colon	1166 (71.4)	172 (71.1)
Rectum	468 (28.6)	70 (28.9)
Histologic differentiation		
Well, moderately	1520 (93.0)	224 (92.6)
Poorly, mucinous, signet-ring cell	53 (3.3)	7 (2.9)
Unknown	61 (3.7)	11 (4.5)
Number of harvested LN, mean ± SD	27 ± 11	28 ± 13
<12 LN harvested	17 (1.2)	5 (2.1)
Lymphovascular invasion	382 (23.4)	26 (23.1)
Perineural invasion	269 (16.5)	31 (12.8)
Preoperative obstruction	437 (26.7)	82 (33.9)
Involved resection margin	22 (1.3)	6 (2.5)
MSI status		
MSS	1292 (79.1)	199 (82.2)
MSI-L	62 (3.8)	0
MSI-H	169 (10.3)	27 (11.2)
Unknown	111 (6.8)	16 (6.6)
Addition of adjuvant chemotherapy	772 (47.2)	127 (52.5)
Follow up duration, mean ± SD, months	59 ± 23	61 ± 23

LN, lymph nodes; MSI, microsatellite instability; MSI-L, MSI low frequency; MSI-H, MSI high frequency; MSS, microsatellite stable; SD, standard deviation.

**Table 2 cancers-14-02833-t002:** Clinicopathologic characteristics of patients with pT3N0 colorectal cancer according to the addition of adjuvant chemotherapy in the overall cohort.

	Adjuvant Chemotherapy (*n* = 772)	No Adjuvant Chemotherapy (*n* = 853)	*p* Value
Age, mean ± SD, years	59 ± 10	67 ± 11	<0.001
Sex			
Male	468 (60.7)	496 (58.2)	
Female	304 (39.3)	357 (41.8)	
Location			<0.001
Colon	492 (63.7)	664 (77.8)	
Rectum	280 (36.3)	189 (22.2)	
Poorly differentiated	29 (3.8)	20 (2.3)	0.116
<12 LN harvested	6 (0.8)	11 (1.3)	0.311
Lymphovascular invasion	280 (36.3)	104 (12.2)	<0.001
Perineural invasion	176 (22.8)	95 (11.1)	<0.001
Preoperative obstruction	236 (30.6)	199 (23.3)	0.001
Involved resection margin	12 (1.6)	10 (1.2)	0.506
MSI-H	73 (9.5)	95 (11.1)	0.128
No. of high-risk features			<0.001
No	245 (31.7)	528 (61.9)	
1	351 (45.5)	238 (27.9)	
≥2	176 (22.8)	87 (10.2)	

LN, lymph nodes; MSI-H, microsatellite high frequency; No., number; SD, standard deviation.

**Table 3 cancers-14-02833-t003:** Factors associated with RFS and OS in the overall cohort of patients with T3N0 colorectal cancer.

Variables	RFS					OS				
	Univariate		Multivariate			Univariate		Multivariate		
	HR	*p*-Value	HR	95% CI	*p* Value	HR	*p* Value	HR	95% CI	*p* Value
Age	1.026	0.003	1.203	1.0078–1.041	0.006	1.096	<0.001	1.086	1.066–1.06	<0.001
Adjuvant chemotherapy	0.932	0.593				0.518	<0.001	0.808	0.576–1.134	0.218
LVI	1.49	0.067	1.157	0.762–1.756	0.494	1.191	0.341			
PNI	2.380	<0.001	2.737	1.827–4.099	<0.001	2.254	<0.001	2.136	1.499–3.043	<0.001
<12 LN harvested	0.860	0.990				2.892	0.036	2.095	0.772–5.881	0.146
Obstruction	1.666	0.008	1.602	1.096–2.343	0.015	1.673	0.002	1.580	1.133–2.203	0.007
PD	0.999	0.987				1.091	0.011	1.099	1.027–1.176	0.006
Margin (+)	4.705	0.001	5.399	2.176–13.399	<0.001	2.723	0.048	3.1	1.139–8.439	0.027

CI, confidence interval; HR, hazard ratio; LN, lymph node; LVI, lymphovascular invasion; obstruction, preoperative obstruction; OS, overall survival, PNI, perineural invasion; PD, poorly differentiated; RFS, recurrence-free survival; Margin (+), resection margin involved.

**Table 4 cancers-14-02833-t004:** Changes in the diagnosis of LVI and PNI in the review cohort.

		Before the Review		Total, *n* (%)
		LVI−	LVI+	
rLVI (SVI + LVI + LaVI) after the review	rLVI−	118 (48.8)	15 (6.2)	133 (55)
	rLVI+	67 (27.7)	42 (17.4)	109 (45)
rPNI after the review		**PNI−**	**PNI+**	
	rPNI−	154 (63.6)	5 (2.1)	159 (65.7)
	rPNI+	56 (23.1)	27 (11.2)	83 (34.3)

LaVI, large vessel invasion; LI, lymphatic invasion; LVI, lymphovascular invasion; rLVI, LVI after review; PNI, perineural invasion; SVI, small vessel invasion.

**Table 5 cancers-14-02833-t005:** Factors associated with RFS and OS in the review cohort.

Variables	RFS		OS				
	Univariate		Univariate		Multivariate		
	HR	*p* Value	HR	*p* Value	HR	95% CI	*p* Value
Age	1.017	0.503	1.074	0.002	1.074	1.026–1.123	0.002
Adjuvant chemotherapy	1.052	0.814	0.749	0.467			
rLVI	3.859	0.043	1.520	0.377			
rPNI	1.958	0.245	3.015	0.022	2.579	0.994–6.694	0.051
<12 LN harvested	0.048	0.759	0.049	0.714			
Obstruction	1.519	0.477	2.953	0.023	3.412	327–8.772	0.011
PD	0.668	0.635	1.023	0.846			
Margin (+)	3.279	0.25	2.332	0.411			

CI, confidence interval; HR, hazard ratio; LN, lymph node; Margin (+), resection margin involved; Obstruction, preoperative obstruction; OS, overall survival; PD, poorly differentiated; RFS, recurrence-free survival; rLVI, reviewed lymphovascular invasion; rPNI, reviewed perineural invasion.

## Data Availability

The data presented in this study are available on request from the corresponding author. The data are not publicly available due to conditions of the ethics committee of our university.

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
