# Peer review of "The Prognostic Reliability of Lymphovascular Invasion for Patients with T3N0 Colorectal Cancer in Adjuvant Chemotherapy Decision Making"

_cancers, 2022, doi:10.3390/cancers14122833_

Round 1
Reviewer 1 Report
The central queestion is to establish whether LVI is a good indicator for adjuvant chemotherapy.
Originality is average however it can improve diagnosis in the field.
Lee et al. proposes an intriguing new prognostic factor for colon cancer, congratulations to the authors for a well-rounded study, minor rephrasing or just spelling errors to correct.
Author Response
I appreciate your comment. Our manuscript has been edited and proofread by a native English-speaking medical editor and numerous editing changes have been made. Thank you for your effort and time taken to review our newly revised manuscript.

Reviewer 2 Report
The study reported in the manuscript challenges the use of LVI status to stratify adjuvant therapy decision making in TN30 colorectal cancer patients. The findings are of high potential importance. Clinical cohort was sufficiently large and the study was conducted with sufficient scientific rigor. However, several aspects of presentation need to be improved before the study could be recommended for publication. Specifically:- While the quality of English is sufficient to understand the ideas that authors conveyed, multiple grammatical errors and questionable phrasing made it unnecessary difficult to follow the study. Just one of the examples "This poor adherence to guideline for adjuvant chemotherapy might be caused that oncologic benefit of adjuvant chemotherapy has not been established yet". The paper would greatly benefit from meticulous editing of both style and grammar.
- Figure 2 that shows a representative image for dual IHC for more accurate diagnosis of LVI. I found the description to be overly cryptic to adequately understand what is shown. More elaboration needed for non-pathologists. Magnification/resolution appear to be insufficient to see the features of relevance; examples of positivity for both CD31 and D2-40 need to be clearly pointed out). Given that the paper makes a point of changing diagnosis after the IHC based re-evaluation, it would have been useful to show clear examples when status was changed after the review.
- While the manuscript nicely explains the reason behind the variability in the diagnosis of LVI, there are no speculations presented for the potential reasons of differences in the PNI diagnosis.
- Details of the IHC staining (antibodies, and protocol) need to be included. Authors need to list chemotherapies that were administered to the patients and describe whether all of the patients received the same chemotherapies.
- Since the paper challenges the diagnostic utility of LVI and PNI status, I would expect to have a more substantial discussion for the causes of this discrepancy. Authors list the variability in pathologists assessment, but more thorough IHC based reevaluation did not change the results. If the study results are representative of the general picture, this mean that there was something wrong with the original reasoning for including LVI and PNI status for diagnostic decision making. I think that the discussion needs to cover that.
Author Response
(Reviewer 2)
The study reported in the manuscript challenges the use of LVI status to stratify adjuvant therapy decision making in TN30 colorectal cancer patients. The findings are of high potential importance. Clinical cohort was sufficiently large and the study was conducted with sufficient scientific rigor. However, several aspects of presentation need to be improved before the study could be recommended for publication. Specifically:
While the quality of English is sufficient to understand the ideas that authors conveyed, multiple grammatical errors and questionable phrasing made it unnecessary difficult to follow the study. Just one of the examples "This poor adherence to guideline for adjuvant chemotherapy might be caused that oncologic benefit of adjuvant chemotherapy has not been established yet". The paper would greatly benefit from meticulous editing of both style and grammar.
Response: Thank you for your comments, which are greatly appreciated. Our manuscript has been extensively revised and edited by a professional editing service. We consider the revised version is greatly improved and we hope that it will now be of a suitable standard for publication in “Cancers”.
Figure 2 that shows a representative image for dual IHC for more accurate diagnosis of LVI. I found the description to be overly cryptic to adequately understand what is shown. More elaboration needed for non-pathologists. Magnification/resolution appear to be insufficient to see the features of relevance; examples of positivity for both CD31 and D2-40 need to be clearly pointed out). Given that the paper makes a point of changing diagnosis after the IHC based re-evaluation, it would have been useful to show clear examples when status was changed after the review.
Response: Thank you for your comment. I agree with your points. For better understanding for non-pathology readers, we have changed Figure 2 to higher magnification images in our revised manuscript as follows:
“Figure 2. Representative images of venous and lymphatic invasion with original H&E staining and dual CD31-D2-40 immunolabeling. (a) Identifying lymphatic or venous invasion on H&E staining is extremely difficult due to retraction artifact (asterisks) making spaces surrounding cancer cells. (b) Dual CD31 (magenta) -D2-40 (brown) immunolabeling highlights cancer cell infiltration in lymphatic space (arrows) among numerous retraction artifact (asterisks). (c) identifying venous invasion on H&E staining is extremely difficult in this case. Unpaired artery (asterisk) is seen. (d) Dual CD31 (magenta) -D2-40 (brown) immunolabeling highlights venous invasion (arrows) of cancer cells. Venous endothelial cells show magenta color (CD31 labeling). (magnification of all images, x200).”
While the manuscript nicely explains the reason behind the variability in the diagnosis of LVI, there are no speculations presented for the potential reasons of differences in the PNI diagnosis.
Response: Under-detection of PNI on the original pathology diagnosis could be explained by inter-observer disagreement concerning the definition of PNI. Therefore, establishing a precise definition of PNI for colorectal cancers may be required.
Details of the IHC staining (antibodies, and protocol) need to be included. Authors need to list chemotherapies that were administered to the patients and describe whether all of the patients received the same chemotherapies.
Response: As the reviewer has suggested, the following description has been added:
“Briefly, 4 μm thick tissue sections were deparaffinized and rehydrated by immersion in xylene and a graded ethanol series. Endogenous peroxidase was blocked by incubation in 3% H2O2 for 10 minutes, followed by heat-induced antigen retrieval. Immunohistochemical labeling was performed with an autostainer (Benchmark XT; Ventana Medical Systems, Tucson, AZ, USA) following to the manufacturer’s protocol. In brief, sections were incubated at room temperature for 32 minutes with primary antibody against D2-40 (M3619, 1:100; Dako, Glostrup, Denmark) and then washed. Additional AP-conjugated mouse monoclonal CD31 (EP78, 1:800; Cell Marque, CA, USA) was used for dual CD31–D2-40 labeling. An ultraView AP Magenta Detection Kit (Ventana Medical Systems) was used for magenta chromogen, and the OptiView DAB Detection Kit (Ventana Medical Systems) was used for the brown chromogen. Immunolabeled sections were lightly counterstained with hematoxylin, dehydrated in ethanol, and cleared in xylene. The immunolabeled slides were reviewed by two pathologists. As a result, CD31 was labeled with magenta color and D2-40 was labeled with brown color, respectively. When tumor cells were surrounded both by CD31 (red labeling) labeled endothelial cells, the invasion was considered as venous invasion (Figure 2D). Lymphatic invasion was confirmed by presence of tumor cells within the lymphatic space surrounded by D2-40 positive lymphatic endothelia (brown labeling; Figure 2B).”
Since the paper challenges the diagnostic utility of LVI and PNI status, I would expect to have a more substantial discussion for the causes of this discrepancy. Authors list the variability in pathologists assessment, but more thorough IHC based reevaluation did not change the results. If the study results are representative of the general picture, this mean that there was something wrong with the original reasoning for including LVI and PNI status for diagnostic decision making. I think that the discussion needs to cover that.
Response: The following description has been added to the Discussion section in our revised manuscript:
“The original pathologic diagnosis was made only with H&E staining slides. Therefore, it might have been difficult to differentiate between retraction artifact and LVI (Figure 2A). In this study, two gastrointestinal pathologists performed a central pathology review using additional dual CD31-D2-40 immunolabeling, and detected more cases with LVI present. Our results are similar to those of several previous studies that have reported detection of more foci of venous or lymphatic invasion using additional CD31 or D2-40 IHC staining, respectively, compared with conventional H&E staining only [14, 30].”

Reviewer 3 Report
- This is an interesting article, which attempts to address a very important topic regarding how-to make a decision about therapeutic regimen in stage III CRC patients.
- Please revise the English language of the manuscript, it needs a careful and thorough review.
- Line 25: Stating or staining?
- Simple summary: Please provide at least some details of the study - mention design, number of patients and length of follow-up.
- Line 27: Was not associated is correct way of stating this, rather than saying associated factor.
- Line 29: This sentence needs to be revised - it is not clear enough to say that importance was not defined. There either was or was not a significant association between LVI and prognosis and survival, which in turn affects the choice of therapy.
- Line 25: In 82 what? Patients.
- Introduction: This section needs to be reorganized. Start off the section with an overview of the problem - describe incidence and mortality from colorectal cancer, describe burden of stage III colorectal cancer, its prognosis and survival. Then describe treatment options and how these decisions are guided. Then explain the evidence gap - why you conducted this study.
- Lines 78-82: Clearly define the aim of this study - this paragraph needs to be revised for clarity.
- Line 85: Clearly define study design.
- Lines 89-90: What do "accurate pathologic results" mean? How can patients have incorrect results? How did you ascertain correctness?
- Line 108: Add definition of overall survival.
- Methods: What was the range of follow-up?
- Table 1: Check results - some values do not add up to 100%, some values are not written correctly, have different decimal spots etc.
- Results: Try revising this section so that less information is repetitive with illustrations, and emphasize the important findings.
- Discussion: Start off this section with a summary of your findings.
- Line 344: Why were you unable to review those before 2014?
- Conclusions: Revise this section - avoid numerical values, provide brief and on-point description of your results and their implication for practice.
Author Response
(Reviewer 3)
This is an interesting article, which attempts to address a very important topic regarding how-to make a decision about therapeutic regimen in stage III CRC patients.
Please revise the English language of the manuscript, it needs a careful and thorough review.
Response: Thank you for your efforts and for the time taken to review our manuscript. We evaluated the reliability of LVI as a high-risk factor when deciding whether to administer adjuvant chemotherapy for patients with pT3N0 colorectal cancer. Our manuscript has been carefully and extensively revised by a native English-speaking medical editor.
Line 25: Stating or staining?
Simple summary: Please provide at least some details of the study - mention design, number of patients and length of follow-up.
Line 27: Was not associated is correct way of stating this, rather than saying associated factor.
Line 29: This sentence needs to be revised - it is not clear enough to say that importance was not defined. There either was or was not a significant association between LVI and prognosis and survival, which in turn affects the choice of therapy.
Line 25: In 82 what? Patients.
Response: Thank you kindly for these very detaied comments. The summary has been revised in lines 29-40, according to your comments.
Introduction: This section needs to be reorganized. Start off the section with an overview of the problem - describe incidence and mortality from colorectal cancer, describe burden of stage III colorectal cancer, its prognosis and survival. Then describe treatment options and how these decisions are guided. Then explain the evidence gap - why you conducted this study.
Lines 78-82: Clearly define the aim of this study - this paragraph needs to be revised for clarity.
Response: Thank you for comments. The Introduction section has been re-organized and the study aim has been revised for enhanced clarity and precision.
We focused on pT3N0 colorectal cancer, which remains controversial in terms of adjuvant treatment. I have included this comment concerning patients with pT3N0 colorectal cancer as a background justification for the study.
Methods: What was the range of follow-up?
Response: The mean follow-up duration was 50 (1-114) months. This information has been included in the Methods sub-section 1.1
Line 85: Clearly define study design
Response: Thank you for your advice. I have included details of this retrospective study design in the Methods section.
Lines 89-90: What do "accurate pathologic results" mean? How can patients have incorrect results? How did you ascertain correctness?
Response: I apologize for this vague expression. I meant patients who could not be accessed pathologically with “accurate pathologic results.” For example, patients with pTx or pNx. As you have pointed out, “accurate pathologic results” is an expression that is open to misinterpretation; therefore, I have changed this to “patients whose pathologic stage could not be assessed.”
Line 108: Add definition of overall survival.
Response: I have added definition of overall survival in lines 133-135.
Results: Try revising this section so that less information is repetitive with illustrations, and emphasize the important findings.
Response: Thank you for your comments. I have revised the Results section to present the important study findings more clearly.
Table 1: Check results - some values do not add up to 100%, some values are not written correctly, have different decimal spots etc.
Response: I have carefully checked Table 1 and have revised it accordingly.
Discussion: Start off this section with a summary of your findings.
Response: Table 1 has been revised in accordance with your suggestion.
Line 344: Why were you unable to review those before 2014?
Response: Thank you for your comment. We wanted to determine whether the prognostic importance of LVI differed according to diagnostic conversion via central review and additional staining. If we had been able to perform a central review for all patients included in this study, it would have been ideal; however, we could not perform a central review of all cases included in this study because of a lack of funding and pathologists due to the large number of patients in the overall cohort (n = 1634). Therefore, we chose to evaluate a review
cohort over one entire year.
Conclusions: Revise this section - avoid numerical values, provide brief and on-point description of your results and their implication for practice.
Response: The conclusion has been revised, in keeping with your suggestions.

Round 2
Reviewer 3 Report
- I would like to thank the Authors for addressing most of my comments.
- Please revise the following:
- Lines 97-99: Consider omitting this sentence, since the previous one describes the aim adequately.
- Table 1: I have to repeat this comment - please check that all percentages add up to 100%, for example column for review cohort for sex - percentages add up to 100.1%, or for histologic differentiation
Author Response
Lines 97-99: Consider omitting this sentence, since the previous one describes the aim adequately.
(Response) I agree with your opinion. I omit following sentence "Two gastrointestinal pathologists assessed the diagnostic reliability of LVI through additionally undertaking central pathologic slide reviews using dual D2-40 and CD31 IHC staining"
Table 1: I have to repeat this comment - please check that all percentages add up to 100%, for example column for review cohort for sex - percentages add up to 100.1%, or for histologic differentiation.
(Response) We are very sorry for mistake and really appreciate your detailed comment and patience. We checked number and sum of it very carefully. We found out sex and histologic differentiation percentages were not correct and revise these. We also checked carefully other numbers.